# Grafting and Soil with Drought Stress Can Increase the Antioxidant Status in Cucumber

**Marcelino Cabrera De la Fuente** [1], **Jesus Tomas Felix Leyva** [2], **Rafael Delgado Martinez** [3],
**Julia Medrano Macías** [1] **and Rocio Maricela Peralta Manjarrez** [4,*]

1 Departament of Horticulture, Universidad Autónoma Agraria Antonio Narro,
Saltillo 25315, Coahuila, Mexico
2 Estudiante Mestría en Ciencias en Horticultura, Universidad Autónoma Agraria Antonio Narro,
Saltillo 25315, Coahuila, Mexico
3 Faculty of Engineering and Sciences, Autonomous University of Tamaulipas,
Ciudad Victoria 87000, Tamaulipas, Mexico
4 Postdoctorante CONACYT, Departamento de Horticultura Universidad Autónoma Agraria Antonio Narro,
Saltillo 25315, Coahuila, Mexico
* Correspondence: rperaltam@hotmail.com; Tel.: +52-844-4110303

**Abstract:** The availability of water and nutrients are determining factors for the growth and productivity of the cucumber crop. The implementation of the grafting techniques increases the efficiency in the absorption of resources such as water and nutrients, improving the quality, mineral content, and enzymatic activity of the fruit. The objective of this research work was to evaluate the effect of the anchor point (soil and substrate), graft (with and without graft), and irrigation volume (100 and 75%) on fruit quality, macro and microelement content, and enzymatic antioxidant activity. A total of eight treatments was established in a completely randomized experiment with a factorial design with a $2 \times 2 \times 2$ arrangement. The results showed an increase in the fruit weight by 10% in grafted plants under 100% irrigation in the substrate, no significant differences between treatments were found in firmness or total soluble solids (TSS). Additionally, while there was a higher accumulation of potassium because of the graft, there was no difference for calcium. It was observed that the enzymatic activity was inhibited using the graft. The graft represents a viable option for the efficient use of water, both in the soil and substrate, being the substrate with the best environment for development, mitigating stress by decreasing enzymatic activity.

**Keywords:** graft cucumber; antioxidants status; commercial quality; irrigation; soil; substrate

## 1. Introduction

Cucumber (*Cucumis sativus* L.) is an economically important vegetable due to its high production worldwide and because it can be consumed both fresh and processed [1]. They are native to Asia, and hence they are widely cultivated in tropical and subtropical regions of the world. In Europe, 26% of the total cucumber is produced, while Mexico is the main producer in Latin America and the seventh in the world, reaching more than 1,000,000 t in 2020 [1].

Its fruits have a wide variety of sizes ranging from 5 to 60 cm long, and colors such as green, creamy, white, yellow, brown, orange, or red [2]. They are beneficial for human health due to their high water content (between 80 and 90%) and bio-compounds such as vitamins C, A, B complex, cucurbitacin, and lignans, among others [3,4].

Due to its high-water content, cucumber crop requires great irrigation, since it interferes with the normal physiological functions of the plants if it is not adequately supplied, causing water stress and damaging the plants at a physiological, biochemical, genetic, and epigenetic level. For that reason water deficit decreases the size of the plant, the foliar area, the yield, and the quality of the crop [5,6].

However, there are many parts of the world that currently have problems with the availability of water for agriculture, mainly due to climate change and human overpopulation, and this problem is expected to worsen in the coming years [7].

Several techniques have been used to deal with water stress in plants. One of the most promising and efficient methods is the grafting technique, which consists of vegetative propagation of two plants are combined to grow as one; the lower part is called rootstock, which is selected to fight against abiotic stress, and the upper part is called the scion, which is chosen based on the quality and yield of fruit required [2].

Furthermore, the grafting technique is considered a quick and easy ecological technique, since it contributes to making water use more efficient, promoting yield and quality in crops [8]. Rootstocks can enhance the uptake of minerals in the grafted plants, due to an improvement in the size and architecture of the roots, although the final result of the elemental concentration also depends on transport, accumulation, recirculation, and growth. To date, some studies have evidenced an increase in macronutrient contents such as N, K, Ca, and Mg with grafting vegetables [3].

In addition, they can contribute to sustainable agriculture by reducing the amount of agrochemicals used to disinfect the soil, since it has been used to induce tolerance to pests and diseases [9].

Another alternative to make water efficient in agriculture is through deficit irrigation (DI), which is defined as a water management method that regulates its supply by reducing its total requirement for optimal crop growth over a period of specific growth. This creates a controlled soil water deficit condition and reduces actual evapotranspiration (ETa) while improving plant WUE [10].

An abiotic stress, such as water deficiency, causes an overproduction of reactive oxygen species (ROS) such as superoxide radicals, peroxide, singlet oxygen, and hydroxyl, among others, which trigger the synthesis of both enzymatic and non-enzymatic antioxidants in order to neutralize them. Within the primary line of defense are enzymes such as superoxide dismutase (SOD), catalase (CAT), ascorbate (APX), and glutathione peroxidase (GPX) [4]. A efficient indicator of the degree of stress is the activity of antioxidant enzymes as well as the mineral content.

Therefore, the objective of this research work was to test the effect of the interaction of grafted cucumber plants grown in either substrate or soil, with different irrigation regimes on the fruit's quality, macro and microelement content, and enzymatic antioxidant activity.

## 2. Materials and Methods

### 2.1. Experimental Site

The experiment was established in the facilities of the Antonio Narro Autonomous Agrarian University at 25°21′22.51″ north latitude and 101°2′9.88″ west longitude, in Coahuila, Mexico, with an altitude of 1760 m above sea level. In a greenhouse with a plastic cover, in addition to having natural ventilation with an oscillating temperature of 36.5 °C as a maximum and a minimum of 8 °C, in addition to a relative humidity of 75% and a greenhouse solar radiation of 4.9 w/m$^2$, the crop was established from 14 May to 30 August 2021.

### 2.2. Plant Material for Graft and Rootstock

Cucumber grafts (Poniente F1, Enza Zaden, Enkhuzen, The Netherlands) were made on a wild pumpkin rootstock (*Cucurbita maxima* × *Cucurbita moschata*), with the rootstock seed placed in 60-cavity polystyrene trays on 7 April 2021. The scion was sown in another polystyrene tray on 9 April 2021, with a peat moss substrate with perlite in a 1:1 ratio.

### 2.3. Making of the Graft

The graft was made on 1 May 2021 using the slit graft method, which consists of making an incision with a sterile razor, immersing the razor in each cut of the rootstock from the aerial basal, which splits into a "Y" shape 2 cm in the middle. The scion is cut by

thinning the stem in such a way that it enters the graft holder. Once this is done, a clamp is placed on it, whose function is to hold the place where the incision was made. Once this is finished, we proceed to place it in a healing chamber. For both cases, the seedlings must have true leaves [11,12].

### 2.4. Healing Chamber and Acclimatization

The survival of the graft depends on an adequate acclimatization due to the healing of the incision made and the hardening of the area. For this, it was necessary to put the plant in a healing chamber, which consists of a transparent membrane to maintain humidity. This transparent membrane is covered by a black plastic cover to keep out the light. It should be noted that the weather situation affects the adhesion of the incision made as well as humidity [11]. The internal conditions of the ideal healing chamber were between 85 and 95% relative humidity as well as a temperature from 20 to 28 °C [9,13–15].

In the healing chamber, the plants lasted 8 days for their acclimatization, in which the transparent membrane was moistened only every two hours in the first three days. During the fourth day, one side of the dark plastic cover was lifted so that the plant could adapt. In sunlight, on the fifth day, perforations were made in the membrane so that the plant could better adapt to the loss of moisture. During the sixth day, more perforations were made to the membrane for better ventilation. On the seventh day, the other side of the dark cover was lifted, and more perforations were made. On the eighth day more perforations were made, and the plants were removed from the chamber on the ninth day.

### 2.5. Experimental Design

The experimental design used was a completely randomized design with a factorial arrangement (2 × 2 × 2), the first factor being the plant with or without grafting, and the second factor being the soil and substrate (75:25 ratio of peat moss and perlite), and the third factor consisted of applying two levels of irrigation (100% and 75%). The irrigation level was determined with a tensiometer for each anchorage site, managing tensions of 20 centibars, lowering it to 14 centibars using the volume of water required as optimal, and 25% was subtracted from this, according to the treatments (Table 1). These treated plants were placed in black plastic bags with eight liters of soil or substrate. In addition, daily irrigations were carried out that ranged from 1 to 5 L during the entire cultivation process. In addition, the Steiner solution was used for mineral nutrition [16]. It was applied daily, according to the phenological phase, through an automatized system. The fertilizers' concentrations were the following: $Ca(NO_3)_2$ $4 \cdot H_2O$—4.5 mmol; $KH_2PO_4$—1 mmol; $MgSO_4 7H_2O$—2 mmol; $KNO_3$—3 mmol; $K_2SO_4$—1.5 mmol; Fe chelated—3 $mgL^{-1}$; $H_3BO_3$—0.5 $mgL^{-1}$; $MnSO_4$—0.7 $mgL^{-1}$; $ZnSO_4$—0.09 mg $L^{-1}$; and $CuSO_4$—0.02 $mgL^{-1}$. The nutrient solution was maintained at pH 6.0 and had an electrical conductivity (EC) of 1.8 dS × $m^{-1}$.

**Table 1.** Treatments used in the Experiment.

| Treatments | Specification |
|---|---|
| 1 | Without Graft–Soil irrigation 100% |
| 2 | Without Graft–Soil irrigation 75% |
| 3 | Without Graft–Substrate irrigation 100% |
| 4 | Without Graft–Substrate irrigation 75% |
| 5 | Graft–Soil irrigation 100% |
| 6 | Graft–Soil irrigation 75% |
| 7 | Graft–Substrate irrigation 100% |
| 8 | Graft–Substrate irrigation 75% |

### 2.6. Average Weight of Cucumber Fruits

For the average value of fruits, the weight of 14 fruits of each treatment was averaged, and this was weighed on a digital scale (model spx2202) with its weight in grams from the company Ohaus (Parsippany, NJ, USA).

### 2.7. Fruits Firmness

Fruit firmness was measured with a penetrometer model Ft 327 from the company QA Supplies (Norfolk, VA, USA), which measures on a kg cm$^{-2}$ scale.

### 2.8. Total Soluble Solids

It was measured with a VeeGee refractometer model 43,003 from the manufacturer Midland Scientific (Omaha, NE, USA) with a °Brix scale, which was quantified in freshly cut fruits.

### 2.9. Minerals

The content of macromineral elements was carried out at the Faculty of Engineering and Sciences in the soil laboratory of the Autonomous University of Tamaulipas, and results were expressed in % (g/100 g dry biomass).

The Nitrogen (N) content was determined by the Kjeldahl method [17], for which 500 mg of dry sample were weighed in a digestion flask, to which 4 mL of digester mixture (25 g of $K_2SO_4$, 10 g of HgO, 1 L of $H_2SO_4$ concentrated, and 25 mL of $Cu_2SO_4$) was added and subjected to the digestion process for 2 h using selenium reactive mixture as catalyst. Subsequently, it was allowed to cool and was subjected to distillation adding 50% NaOH. The distillation was recovered with 30 mL of 2.2% boric acid and 3–5 drops of bromocresol green and methyl red mixed indicator. Then, an assessment was made by titration with $H_2SO_4$ 0.025 N.

Calcium (Ca) and Potassium (K) quantification was performed by acid digestion with nitric acid ($HNO_3$), using 500 mg of fruit dry matter, which was placed in Teflon containers with 10 mL of acid and digested in a MARS 6 microwave digestion system for 1 h. Elemental quantification was performed using an Inductively Coupled Plasma (ICP) emission spectrophotometer of the Termo Jarrel Ash Irish Advantage model.

The Phosphorus (P) content was determined by the colorimetric technique in UV-Visible spectrometry [18], using a heptamolybdate-vanadate solution at an absorbance of 470, and the results were determined using the calibration standard 20 mg L$^{-1}$ of P.

### 2.10. Extraction of Biomolecules for Enzymatic Activity

The plant material was lyophilized and, subsequently, macerated with a pestle until a fine powder was obtained, of which 100 mg were taken and added with 10 mg of polyvinyl pyrrolidone (pvp) and placed in a 2 mL microtube. The extraction was carried out by adding 2 mL of phosphate buffer pH 7–7.2 (0.1 M). It was homogenized and subjected to sonication for 5 min. Subsequently, centrifugation was carried out at 12,500 revolutions per minute (rpm) for 10 min at 4 °C, the supernatant was collected and filtered with a 0.45-micrometer nylon membrane, and it was, finally, diluted in a proportion of 1:15 with phosphate buffer pH 7–7.2 (0.1 M).

### 2.11. Total Proteins

Protein determination was performed by the Bradford method [19]. A total of 0.1 mL of extract was taken, and 1 mL of Bradford's reagent was added. The mixture was incubated for 10 min, and the wavelength was measured at 595 nm in a UV-VIS spectrophotometer using a plastic cell. The values obtained were reported in grams per kilogram of dry weight of plant material (g kg$^{-1}$ DW).

### 2.12. Catalase EC 1.11.1.6

The quantification of the catalase activity was carried out at time zero and time one, which were read in the UV-VIS spectrophotometer at a wavelength of 270 nm in a quartz cell, according to the methodology of [20]. For time zero, 0.1 mL of sample was added in a micro tube, to which 0.4 mL of 5% trichloracetic acid was placed with, later, 1 mL of 100 mM hydrogen peroxide. For time 1, 0.1 mL of sample was added, 1 mL of 100 mM hydrogen peroxide was added, and the mixture was stirred for one minute, after which 0.4 mL of 5% trichloroacetic acid was added to stop the reaction. Catalase activity was reported as U per gram of total protein (U $g^{-1}$ TP), where U equaled mM $H_2O_2$ consumed per milliliter per minute.

### 2.13. Ascorbate Peroxidase EC 1.11.1.11

The measurement of the enzymatic activity ascorbate peroxidase was carried out, according to what was established by Nakano and Asada [21]. A difference of two times was made in time zero and time one, where, for time zero, 0.1 mL of the enzyme extract was used in a micro tube, 0.5 mL of ascorbate was added at 10 mg $L^{-1}$, after which 0.4 mL was added of 5% trichloroacetic acid to stop the reaction, and 1 mL of 100 mM hydrogen peroxide was added. For time one, 0.1 mL of the enzyme extract was added in a micro tube with 0.5 mL of ascorbate 10 mg $L^{-1}$ and 1 mL of 100 mM hydrogen peroxide. After one minute of stirring, trichloroacetic acid 5% was added to stop the reaction. Both samples were read in the UV-VIS spectrophotometer at 266 nm in a quartz cell. APX activity was reported as U per gram of total protein (U $g^{-1}$ PT), where U equaled μmol of oxidized ascorbate per milliliter per minute.

### 2.14. Glutathione Peroxidase EC 1.11.1.9

The method modified by Flohé and Günzler, adapted by Xue [22], was used. In a micro tube, 0.2 mL of enzymatic extract, 0.4 mL of glutathione reduced at 0.1 mM, and 0.2 mL of $Na_2HPO_4$ (67 mM) were mixed. Subsequently, 0.2 mL of 1.3 mM hydrogen peroxide was added to start the catalytic reaction. After 10 min, 1 mL of trichloroacetic acid 1% was added to stop the reaction and it was passed through an ice bath for 30 min. Then, it was centrifuged at 3000 rpm at 4 °C for 10 min. For quantification, 0.48 mL of the supernatant was taken and placed in a test tube, 2.2 mL of 0.32 M $Na_2HPO_4$ was added, and 0.32 mL of 1 mM 5.5 dithio-bis-2 nitro benzoic acid (DTNB) dye was added and read in a UV-VIS spectrophotometer at 412 nm with a quartz cell. The results were expressed in U per gram of total protein (U $g^{-1}$ PT), where U was equal to mM GSH per milliliter per minute.

### 2.15. Superoxide Dismutase EC 1.15.1.1

The determination of superoxide dismutase enzymatic activity was carried out using the SOD Cayman 706002$^{®}$ kit. Using 40 μL of extract, 400 μL of radical detector (tetrazolium salt) and 40 μL of xanthine oxidase solution were added. The mixture was incubated at room temperature for 30 min and the absorbance was measured at a length of 450 nm in a UV-VIS spectrophotometer. SOD activity was reported as U per gram of total protein (U $g^{-1}$ PT).

### 2.16. Statistical Analysis

An analysis of variance and comparison of means was made using the Tukey test ($p \leq 0.05$), evaluated in the Infostat 5 2020 software, see Table 1.

## 3. Results

### 3.1. Variables of Commercial Quality of Cucumber Fruit

When the plants were grown in substrate, a significant increase in the average fruit weight was observed, since it increased by 10.01% more in the grafted plants compared to the non-grafted treatment under adequate irrigation conditions (Table 2). The restriction of irrigation was 75% in grafted plants, and the grafted plants that had 100% irrigation

were statistically the same; however, an increase of 24% was observed compared to the non-grafted ones.

**Table 2.** Comparison of means between treatments for commercial variables of cucumber fruit.

| Treatments | Average Fruits Weight (g) | Firmness (kg cm$^{-1}$) | Total Soluble Solids (°Brix) |
|---|---|---|---|
| Without Graft-So-100% | 466.57 cd | 6.37 a | 5.00 a |
| Without Graft-So-75% | 403.79 d | 5.53 a | 5.33 a |
| Without Graft-Sub-100% | 715.86 ab | 6.83 a | 4.67 a |
| Without Graft-Sub-75% | 544.07 cd | 5.10 a | 5.17 a |
| Graft-Soil-100% | 578.21 bc | 6.93 a | 4.67 a |
| Graft-Soil-75% | 499.93 cd | 6.33 a | 4.50 a |
| Graft-Sub-100% | 795.50 a | 5.80 a | 4.00 a |
| Graft-Sub-75% | 715.36 ab | 6.33 a | 4.50 a |
| CV | 23.88 | 23.27 | 17.67 |

Note. Abbreviations: So: Soil, Sub: Substrate, 100%: normal irrigation, and 75%: deficit irrigation; Means with different letters are significantly different, and the comparison test was performed by means of Tukey ($p \leq 0.05$).

On the other hand, when the plants were cultivated in soil, the grafting factor was extremely relevant, since a decrease in the average weight of fruits of 34.82% was observed in non-grafted plants compared to grafted plants in 100% irrigated soil. A similar trend was presented when irrigation was restricted to 75%, since the grafted plants increased by 25.78% more than the crop without grafting (Table 2).

The firmness of the fruit, as well as the content of total soluble solids, did not change with the application of the treatments.

### 3.2. Minerals

There were significant differences between treatments in the content of the macroelements nitrogen, phosphorus, and potassium in the fruits.

Regarding the nitrogen content, the grafting factor was not decisive for its accumulation, but the type of substrate was, since an increase of 9% was observed with the No-Sust-100% treatment compared to No-So-100%, followed by No-Sust-75%, which increased 8.8% more than No-So-75% (Table 3).

**Table 3.** Comparison of means of treatments in the mineral content of the fruits.

| Treatment | Nitrogen % | Phosphorous % | Potassium % | Calcium % |
|---|---|---|---|---|
| Without Graft-So-100% | 1.30 c | 0.43 c | 3.98 ab | 2.42 a |
| Without Graft-So-75% | 1.24 d | 0.37 de | 3.18 c | 2.09 a |
| Without Graft-Sub-100% | 1.43 a | 0.55 b | 3.75 abc | 2.74 a |
| Without Graft-Sub-75% | 1.36 b | 0.54 b | 3.93 ab | 1.85 a |
| Graft-Soil-100% | 1.17 e | 0.35 e | 3.60 bc | 2.59 a |
| Graft-Soil-75% | 1.19 e | 0.39 d | 3.90 ab | 2.68 a |
| Graft-Sub-100% | 1.24 d | 0.52 b | 4.05 ab | 1.73 a |
| Graft-Sub-75% | 1.36 b | 0.60 a | 4.30 a | 2.22 a |
| CV | 1.62 | 3.29 | 6.70 | 23.09 |

Note. Abbreviations: So: Soil, Sub: Substrate, 100%: normal irrigation, and 75%: deficit irrigation; Means with different letters are significantly different, and the comparison test was performed by means of Tukey ($p \leq 0.05$).

Cucumber grafting and the use of substrate and restricting irrigation to 75% caused an increase in phosphorus content of 13% more than the treatment with optimal irrigation W-Sust-100%. In addition, it increased 8.3 and 10% more than the non-grafted treatments No-Sust-100% and No-Sust-75%, respectively (Table 3).

On the other hand, with the cultivation of the plants in soil, the use of graft under the optimal irrigation regime caused a decrease in the phosphorous accumulation of 18%, with respect to the non-grafted treatment No-So-100%.

The percentage of potassium in cucumber fruits increased mainly in the W-Sust-75% treatment with a mean of 4.30, which represented an increase of 5%, with respect to the W-Sust-100% treatment. When the plants were established in soil, it was possible to clearly appreciate the grafting effect, since this element was increased up to 18% more than the No-So-75% treatment (Table 3).

### 3.3. Enzymatic Activity

The activity of the antioxidant enzymes catalase, ascorbate peroxidase, glutathione peroxidase, and superoxide dismutase in cucumber fruits had significant differences between treatments.

The highest catalase activity was obtained in the treatment without grafting in soil with 75% irrigation (No-So-75%), since it increased 47% more than the treatment with optimal irrigation conditions No-So-100% (Table 4). This same tendency was observed in plants grafted on soil, and it was also possible to increase this enzyme up to 56% with the treatment W-So-75% compared to W-So-100%. This makes it clear that the determining factor in the catalase activity was the irrigation regime, regardless of the graft and the substrate used.

**Table 4.** Comparison of means for the treatments in enzymatic activity of the fruits.

| Treatment | Catalase (U g$^{-1}$ PT) | Ascorbate Peroxidase (U g$^{-1}$ PT) | Glutathione Peroxidase (U g$^{-1}$ PT) | Superoxide Dismutase (U g$^{-1}$ PT) |
|---|---|---|---|---|
| Without Graft-So-100% | 1.94 bc | 1.06 a | 4.28 b | 1.19 b |
| Without Graft-So-75% | 3.69 a | 0.56 b | 6.01 a | 2.18 a |
| Without Graft-Sub-100% | 1.29 cd | 0.30 b | 2.93 c | 1.00 bc |
| Without Graft-Sub-75% | 0.80 cd | 0.39 b | 1.79 d | 0.71 bc |
| Graft-Soil-100% | 1.11 cd | 0.28 b | 1.44 d | 0.33 c |
| Graft-Soil-75% | 2.57 ab | 0.30 b | 1.55 d | 0.46 c |
| Graft-Sub-100% | 0.35 d | 0.32 b | 1.29 d | 0.33 c |
| Graft-Sub-75% | 0.65 d | 1.26 a | 1.22 d | 0.35 c |
| CV | 33.41 | 24.04 | 15.69 | 36.38 |

Note. Abbreviations: So: Soil, Sub: Substrate, 100%: normal irrigation and 75%: deficit irrigation; Means with different letters are significantly different, and the comparison test was performed by means of Tukey ($p \leq 0.05$).

On the other hand, the activity of ascorbate peroxidase increased considerably when the plants were grafted on a substrate with 75% irrigation (W-Sub-75%), since the activity of this enzyme increased 74% more than W-Sust-100%; it was also superior to the rest of the treatments. However, non-grafted plants established in soil with 100% irrigation also increased 47% more ascorbate peroxidase than non-grafted plants under the 75% irrigation regime (Table 4).

When the cucumber plants were not grafted and established in soil under the 75% irrigation regime, the highest glutathione peroxidase activity was achieved, increasing by 28% with respect to No-So-100%; the rest of the treatments were statistically inferior to No-So-75% (Table 4).

This same trend was observed in the superoxide dismutase activity, since No-So-75% was statistically superior to all treatments, highlighting an increase of 45% more compared to No-So-100% (Table 4).

## 4. Discussion

In the present investigation, a significant increase in the average of fruits weights was obtained when the plants were grafted and cultivated in substrate with 100% irrigation, which suggests an optimal condition for the development of a plant; however, they were

statistically equal to the treatment of grafting with the restriction of irrigation to 75%, which verifies that grafting improved the use of water. [9]. Some studies that have experimented with grafting pumpkin as rootstock have argued that part of the explanation for the increase in efficiency in the use of water was due to the ability of this species to develop.

Root systems have a greater surface area, length, density, and root hairs that allow the better uptake of water. To achieve greater growth, pumpkin is capable of relocating assimilates from its shoot to its roots [5]. Additionally, pumpkin has demonstrated high compatibility with cucumber as a rootstock [9,23,24], with the successful formation of the graft union triggering molecular signaling through the phloem that leads to anatomical and physiological changes in both components for fluid connectivity of vascular tissues, achieving seamless communication between the scion and rootstock of the grafted plant for the mobilization of nutrients, water, and photosynthates [25].

Additionally, it has been documented that it conserves a wide variety of potentially mobile genes that code for the expression of heat shock proteins (HSP70), which are highly conserved proteins involved in responses to water stress [26].

The wild pumpkin has been characterized for also having wide adaptability and great qualities over other varieties, such as resistances to extreme temperature variation and to some types of viruses and diseases of great importance in cucurbits such as powdery mildew [27]. This makes this species a perfect candidate to be used with other varieties of vegetables to face several types of stress.

In the present study, the grafting factor was not determined for the accumulation of nitrogen in cucumber fruits, which is consistent with other investigations carried out in cucurbits, solanaceae, and sapindaceae [28,29]. However, it is possible to observe the increase in nitrogen when the plants are cultivated in substrate with respect to those that are cultivated in soil, since the formulated agricultural substrates have shown to have the physical, physical-chemical, chemical, and biological properties adequate for the optimal growth of crops [30], substantially improving the assimilation of nutrients and, therefore, the quality and agricultural productivity [31].

Cucumber grafting and the use of substrate, restricting irrigation to 75%, caused an increase in the phosphorus and potassium content, inducing plant resistance to water stress. Similar results were observed in research carried out by Aslam et al. (2020) [9], who reported a significant increase in potassium, phosphorus, and magnesium concentrations in Kalam F1 cucumber fruits grafted on pumpkin (*C. pepo*). In the same way, Yuan et al. (2021) [32] reported an increase in phosphorus and potassium, in addition to the microelements iron, manganese, and zinc, when grafting eggplant plants (*Solanum melongena* L.) in Torubamu (*Solanum torvum* L.). The aforementioned authors explained that this physiological phenomenon is also related to the increase in the growth and structure of the rootstock root.

On the other hand, to prevent damage caused by oxidative stress, a balance of reactive oxygen species (ROS) is maintained in plants through multiple pathways and approaches including enzymatic and non-enzymatic defense systems [33]. The enzyme system comprises superoxide dismutase (SOD), catalase (CAT), peroxidase (POD), glutathione peroxidase (GPX), ascorbate peroxidase (APX), monodehydroascorbate reductase (MDHAR), dehydroascorbate reductase (DHAR), and glutathione reductase (GR). These enzymes are responsible for the neutralization of ROS and the maintenance of a redox balance in plant cells [33,34].

Deficit irrigation can contribute to triggering the activation of the antioxidant defense system of plants. Salim et al. (2021) [35] reported that decreasing irrigation to 80% increased the activity of the enzymes superoxide dismutase and catalase in pumpkin plants (*C. pepo* L.g).

The results found in this research work show a reduction in antioxidant enzymes activity; this fact is similar to what was previously reported in [36], which grafted the cucumber hybrid 1010 F1 with four rootstocks: pumpkin (*Cucurbita moschata* L.), bottle



gourd (*Lagenaria siceraria* L.), Nubian watermelon (*Citrullus lanatus* L. var colocynthoide), and winter squash (*Cucurbita maxima* L. var *Flexil*).

The measurement of antioxidant enzymes has been used as indicators of stress relief; hence, the reduction in SOD, CAT, APX, and GPX activity are proportional reductions in reactive oxygen species (ROS), relieving the water deficit by improving water uptake through grafting [6].

## 5. Conclusions

Cucumber grafting and the use of substrate are practical alternatives to make water use more efficient by up to 25%, since the present investigation proves that the average weight of the fruits can be increased, in addition to the phosphorus and potassium content and the activity of the enzyme ascorbate peroxidase in grafted cucumber fruits grown in substrate with a 75% irrigation regime.

Additionally, moderate deficit irrigation was a determining factor in the activation of catalase, ascorbate peroxidase, and superoxide dismutase enzymes, which indicates a better antioxidant defense system that allows plants to remain in an optimum condition to improve their quality and productivity.

**Author Contributions:** Conceptualization, M.C.D.l.F. and R.M.P.M.; methodology, R.D.M.; software, J.T.F.L.; validation, J.M.M. and R.M.P.M.; formal analysis, M.C.D.l.F.; investigation, J.T.F.L.; resources, R.D.M.; data curation, M.C.D.l.F.; writing—original draft preparation, R.M.P.M.; writing—review and editing, M.C.D.l.F.; visualization, J.M.M.; supervision, M.C.D.l.F.; project administration, M.C.D.l.F.; funding acquisition, M.C.D.l.F. All authors have read and agreed to the published version of the manuscript.

**Funding:** This research was funded by University Autonomus Agrarian Antonio Narro, grant number 2462.

**Data Availability Statement:** Not applicable.

**Acknowledgments:** To the supporting research staff at Universidad Autónoma Agraria Antonio Narro and the graduate student scholarship from CONACyT.

**Conflicts of Interest:** The authors declare no conflict of interest.

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
