# Peer review of "Grafting and Soil with Drought Stress Can Increase the Antioxidant Status in Cucumber"

_agronomy, doi:10.3390/agronomy13040994_

Round 1

Reviewer 1 Report

Dear author,

Thanks for your good work. Please see my comments in the attached file.

Regards

Author Response

The iformation was attended.

Reviewer 2 Report

Dear Editor,

The work is relevant, however, it needs changes for future publication. The suggestions are attached below. Some sections should be rewritten to improve the work, especially the discussion, in which the authors do not discuss the results but compare them with other similar results from other studies, and this is not recommended.

With the improvements, the article has the potential to be published in the future.

Tittle

Line 2: Delete the word “type”.

Abstract

Line 18: Put this information in the methodology section, after the objective of the work.

Line 21: The objective is too long. There is no need to describe the analyzed variables. The objective has to be brief and objective.

Line 24: Explore the results further. That seems like a conclusion. Describes the data obtained. Focus only on what was meaningful and had an effect. What didn't make a significant difference isn't as important for your research.

Line 29: Add more keywords

Introduction

Line 42: What decreases? Drought stress?

Line 61: Reduce further. Move this objective to the Abstract.

Material and Methods

Line 69: How was this climate data measured? Did you use any device or weather station?

  Enter that information here.

Line 80: You don't need to quote the date, the year and month is enough.

Line 90: Was this wetting of the membrane done manually?

Line 104: Triple factorial

No need to describe first treatment, second, third...

Put only the factors, factor A, factor B and factor C...

Line 125: Merge subtopics 2.6, 2.7 and 2.8.

Line 211: Wasn't a normality and homoscedasticity test performed before the ANOVA (Analysis of variance)? Review.

Results

Line 213: Every time you describe a result, quote the table or figure at the end of the sentence. Tip for the rest of the text.

Line 215: The table must be inserted after its citation in the text. The order is reversed. Correct this. In Table 1, put % after the CV values. Do this for all tables.

Line 219: This information does not need to be written here. Delete.

Line 255: This information is not required. Just looking at the table we realize that.

Line 262: This information is not required. Just looking at the table we realize that.

Discussion

Line 285: The entire discussion should be rewritten and augmented. The Discussion is based on the comparison of the present study with previous studies, without agreeing on the possible causes for the results found. Review this entire topic.

Liner 286: This is a literature review. Although you included grafting in the study, this information is not directly related to your work. It is general information.

Author Response

The Ntes was attended

Reviewer 3 Report

Line 86. Citations 23 and 32 have nothing to do with what is discussed in point 2.3.

Line 94. There are no authors 60 and 67 in the BIBLIOGRAPHY!

Lines 103-123 (2.3. Experimental design) are difficult to understand, there are no abbreviations like those in the tables for each factor or treatment!

Line 146. The content in N, the result was expressed in…..!?

Line 151. The content in As, the result was expressed in…..!?

Line 163. The content in P, the result was expressed in…..!?

Line 290. Yan et al., (13) in their study mention grafting as very important methods in vegetable cultivation, they do not refer to the quality of the fruits!

Line 297. I suggest changing the term pumpkin crop to pumpkin plants, it is more relevant!

Row 332. Is nitrogen accumulation consistent with the studies of authors 23, 24, 25 or 23 and 25?

This study could benefit from more careful editing by the authors! The citation in the text of references 60 and 67 and the absence from the citations of those from 13 to 23 are unacceptable. The study carried out is interesting, but the way of presentation made it difficult to follow the authors' arguments both in the text and in the tables. Within the tables, I would suggest presenting the variants (8 variants) ungrafted on soil with ungrafted on substrate and grafted on soil with grafted on substrate, forming two groups. It is easier to compare, interpret and understand. I think that the title of the article could be revised in such a way as to highlight the effect of grafting, soil type and lack of water on the nutritional compounds of cucumber fruits.

Author Response

The notes was attended.

Round 2

Reviewer 2 Report

Dear,

Much of what has been suggested has been seen.

Some minor corrections need to be made:

The factors are described incorrectly. At work there are only 3 factors (triple factorial 2x2x2). Fix that and describe them.

The discussion was not restructured, as suggested. Review.

Author Response

Changes Done

Reviewer 3 Report

I have no suggestions for authors!

Author Response

Changes done
